# SynerGPT: In-Context Learning for Personalized Drug Synergy Prediction and Drug Design

## Abstract

Predicting synergistic drug combinations can help accelerate discovery of cancer treatments, particularly therapies personalized to a patient's specific tumor via biopsied cells. In this paper, we propose a novel setting and models for *in-context drug synergy learning*. We are given a small "personalized dataset" of 10-20 drug synergy relationships in the context of specific cancer cell targets. Our goal is to predict additional drug synergy relationships in that context. Inspired by recent work that pre-trains a GPT language model (LM) to "in-context learn" common function classes, we devise novel pre-training schemes that enable a GPT model to in-context learn "drug synergy functions". Our model—which does not use any textual corpora, molecular fingerprints, protein interaction or any other domain-specific knowledge— is able to achieve competitive results. We further integrate our in-context approach with a genetic algorithm to optimize model prompts and select synergy candidates to test after conducting a patient biopsy. Finally, we explore a novel task of inverse drug design which can potentially enable the design of drugs that synergize specifically to target a given patient's "personalized dataset". Our findings can potentially have an important impact on precision cancer medicine, and also raise intriguing questions on non-textual pre-training for LMs.[1]

## 1 Introduction

Drug combination therapy is a standard practice for diseases including cancer (Mokhtari et al., 2017) and HIV. It is based on identifying multiple single agent therapies that, when used together, lead to synergistic effects. Predicting such combinatorial synergies is challenging, especially given the wide range of multiple different mutations as well as different genetic backgrounds typically found in different patients' cancer cells (Mroz & Rocco, 2017). Many drug combinations can also cause increased toxicity (Zapata et al., 2020; Juurlink et al., 2004) in a manner that may depend on specific patient backgrounds (O'Donnell & Dolan, 2009), adding further complexity to the problem. To enable the safest and most effective implementation of combination therapy in cancer care, it is thus important to *personalize* the prediction of drug synergies.

Since the number of drug combinations scales exponentially, differentiating between synergistic and antagonistic pairings is very expensive to test in large quantities in laboratory conditions. Thus, considerable interest has recently grown in using machine learning for predicting synergistic and antagonistic effects between pairs of drugs in silico (Liu et al., 2020; Preuer et al., 2018; Rozemberczki et al., 2022a). These approaches are typically not evaluated in the few-shot setting, where only a few training examples are given. This is particularly relevant in the personalized setting described above, and more generally for cancer tissue types for which there is limited training data for synergy learning models. Additionally, these efforts use a variety of features to categorize the drugs, from molecular fingerprints (Preuer et al., 2018) to protein interactions (Yang et al., 2021). Obtaining these features often requires integrating external knowledge sources (e.g., from drug databases), which often results in findings being restricted to the limited subsets of drugs for which this information is available and also requires specialized engineering in model design. Finally, it is unclear if these external sources are actually needed for current models.

In this work, we address these limitations by exploring the ability of transformer language models (LMs) to learn drug synergy relations. We devise approaches that leverage transformers (1) *without*

---

[1]Code will be made available upon publication.

any external knowledge required to be integrated into the model (i.e., no protein interaction networks or patient cell line features); (2) in the few-shot setting with an in-context learning approach that can generalize to novel unseen drugs and patient cell lines; and (3) for designing novel synergistic drug structures in the context of a specific patient's data.

**Transformer LMs are Strong Drug Synergy Learners—Even Without Textual Representations**
First, we consider drug synergy prediction using transformer language models without enriching drugs/cells with information from external knowledge bases. We find these "feature-less" models are able to achieve results that are better or competitive in comparison to knowledge-enhanced state-of-art drug synergy models (e.g., BERT models achieve 84.1% ROC-AUC to GraphSynergy's 83.4%) Furthermore, in contrast to recent work that uses language models pre-trained on scientific corpora (Nadkarni et al., 2021), we discover an intriguing finding: using *randomized* (i.e. uninformative) tokens instead of drug/cell names is able to rival models that use textual names of those entities. This suggests that external information coming from pre-training on scientific corpora (e.g., as in SciBERT (Beltagy et al., 2019)) or the web (e.g., Wikipedia) has negligible impact on fine tuned models in this setting. These findings motivate us to explore the power of transformer models without external information, and to study generalization beyond memorization capacity by evaluating on novel drugs/cells that were unseen during training.

**SynerGPT: A New In-Context Drug Synergy Setting & Model**    We take inspiration from recent work (Garg et al., 2022) that showed how a GPT model architecture can be trained to "in-context learn" function classes such as linear functions (e.g., linear regression/classification) and neural networks. We pre-train a GPT model from scratch on known drug synergies—using no textual corpora—and explore its ability to generalize in the few-shot setting to drugs and patient cell lines *unseen* during training. We find that our model, dubbed SynerGPT, is able to achieve strong competitive results *without* any external knowledge sources. In particular, we introduce a new setting of *In-Context Learning for Drug Synergy* (ICL-DS). In-Context Learning (ICL) (Dong et al., 2022) has emerged as a powerful paradigm for few-shot learning (Brown et al., 2020). In ICL, trained model parameters are never explicitly updated after pre-training, and adaptation to each task is done on the fly given contextual examples. This is particularly appealing in settings where it is prohibitively costly to perform parameter updates for each incoming new task and context (e.g., for each new patient in a hospital setting). We devise novel pre-training approaches for ICL-DS, including strategies for optimizing the language model prompt selection with a genetic algorithm. Prompts comprise specific combinations of drugs tested for synergy on specific patient cell lines; optimizing prompt selection in this setting has potential implications for the design of a standardized assay panel of drugs and cells to be tested for a patient's particular tumor. While specific patient data at this level is not readily available, we re-purpose existing drug combination data to lay the foundations for formalizing and studying our approaches from a machine learning perspective.

**Designing New Molecules to be Synergistic in the Context of a Specific Patient**    Finally, in our third major contribution we propose an additional new task of *Inverse Synergistic Drug Structure Design* (ISDSD): using a GPT transformer model for *generating* or *retrieving* drug molecules that are synergistic in the context of a specific cancer patient's information (i.e., molecules that are synergistic with other drugs administered to a patient with specific cancer cells). This approach may in the future provide a new methodology for personalized drug candidate discovery.

## 2    BACKGROUND AND PROBLEM SETTING

In the last few decades, combination therapy has emerged as an effective method to target genetically unstable diseases (Mokhtari et al., 2017), with dramatic success in treating HIV (Moore & Chaisson, 1999) and more recently HCV(Liang & Ghany, 2013). Unlike HIV and HCV which encode only 10-15 proteins (Frankel & Young, 1998; Dubuisson, 2007), cancer is radically more complex. Since cancer has an unstable genome, combination therapy is often considered necessary (Mokhtari et al., 2017) and is commonly used in practice, with varying degrees of success.

Generally, drugs work by affecting cellular pathways–chain interactions of molecules which lead to changes in a cell. In *drug synergy prediction*, our goal is to predict whether combining drugs will have positive or negative outcomes in the complex system of these interacting pathways. Generally, synergy

lab experiments are conducted in cell lines, which are a population of cells from a multi-cellular organism (for example, human lung cancer cells). In this work, we also investigate *inverse design* of drug molecules. Traditionally, the idea behind inverse design of molecules is to predict or retrieve a molecular structure which has some desired chemical property or protein target (Sanchez-Lengeling & Aspuru-Guzik, 2018). In our work, we seek to explore inverse design at a higher level– the "interactome" of drug interactions in complex cellular pathways.

**General Problem Formulation** Given $k$ input drugs $d^1, d^2, \ldots, d^k \in \mathcal{D}$ along with a cell line $c \in C$, the goal of drug synergy prediction is to predict a synergy value $y$ for the interactions between the drugs in the given cell line. In existing datasets, only the pairwise $k = 2$ setting is considered. Thus, we focus our experiments on pairwise drug synergy, the most commonly researched setting, but our methods can naturally be extended to n-ary synergies. This problem can be considered as either a regression ($y \in \mathbb{R}$) or a binary classification problem (synergistic (True) or not (False); $y \in [0, 1]$). Synergy data comes from a dataset of tuples $(d^1, d^2, c, y) \in \mathfrak{D}$.

**Few-Shot In-Context Setting** We also consider the few-shot setting in our formulation, which has applications for predicting synergies when there is scarce training data such as in tumor-specific synergy prediction, uncommon cancer tissues, or newly introduced single-agent therapies. In the few-shot setting, we assume there are $n$ synergy tuples available which contain an unknown entity $h$ (unknown cell line $c^h$ or unknown drug $d^h$). Define these tuples as $x_i := (d^1, d^2, c, y)_i$ for $i \in [1..n]$ where one of $d^1$, $d^2$, or $c$ is the unknown $h$. Each $x_i$ can then be used for training in addition to the existing training data. In our proposed method SynerGPT, we don't use these tuples $x_i$ in training– rather, we use them as the prompt for in-context learning. Here, we are particularly interested in synergy prediction based on extremely small datasets (e.g. tested synergies from a patient's specific cancer cells), which makes traditional supervised approaches less effective. In section 3.2.3, we detail our training strategies for in-context learning with unknown $h$ from limited examples.

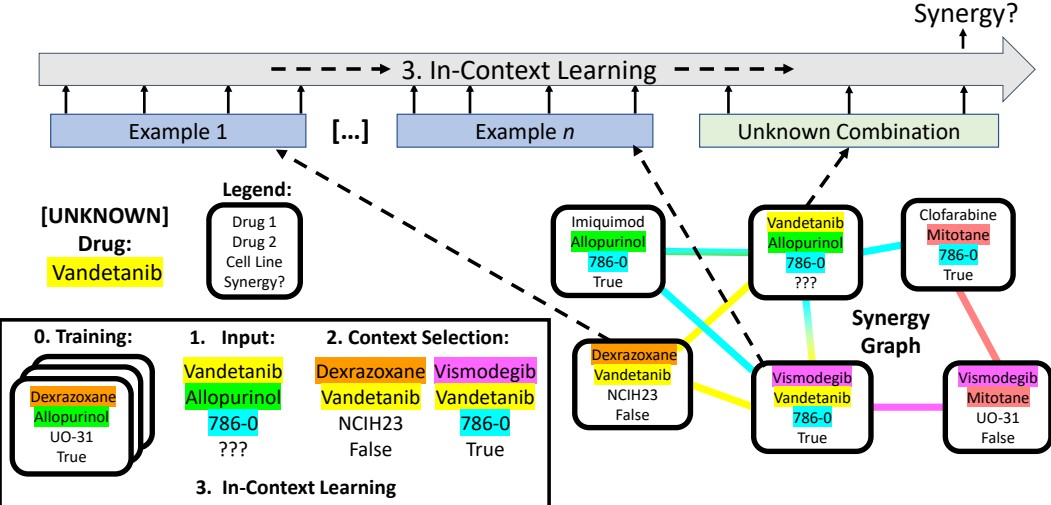

Figure 1: Example of our prompt selection strategy (steps shown in black box). After training (step 0), we are given as input (step 1) a combination between three entities: a known drug Allopurinol, unknown drug Vandetanib, and a known patient cell line 786-0. We are given a small synergy graph $\mathcal{G}$ (bottom right). Nodes represent (drug, drug, cell line) tuples with synergy labels (from previous experiments). Edges represent shared entities; edge color indicates which entities (e.g. red, for sharing Mitotane). Using different strategies, we adaptively select contextual examples (step 2) for in-context learning (step 3).

**Inverse Drug Design from Drug Synergy Context** We propose a new task where the goal is to predict the structure of a molecule given a context of drug synergy tuples (e.g., we might be given 20 synergy tuples). We train a model to predict the structure of some unknown drug $d^h$ from its synergy relations with other drugs. This has two important uses. First, this may enable scientists to predict new molecules which have desirable synergies or similar synergies to existing drugs, which is

a novel way to consider drug discovery. This can potentially enable the design of drugs that synergize specifically to target a given patient's unique cancer cells. Secondly, this can support explainability of the synergy prediction model as a function of the context it is fed, by "visualizing" SynerGPT's understanding of the unknown drug given the context. Section 4.3 shows that we can observe the structure of the drug evolving towards the ground truth as more context examples are given. As this is a novel and difficult problem; we initially frame it as a retrieval task, effectively constraining the output space, though from an implementation perspective it is trivial to instead predict structures by using a pretrained generative model for molecules (Jin et al., 2020) with no architectural differences, as both the retrieval and generation of drug structures requires generating a latent vector.

## 3 Methodology

In this section, we will consider the four components of our paper. First, we detail how drug synergy tuples are input to encoder-only language models (§ 3.1). Next, we extend this idea to the few-shot ICL setting and propose training methodologies to do so (§ 3.2). We then discuss optimization of the "prompt" used for ICL (§ 3.2.3). Finally, we extend our methodology to inverse drug design (§ 3.3).

### 3.1 Input for encoder-only language models

Initially, we explore the efficacy of BERT-style language models (Devlin et al., 2019; Beltagy et al., 2019; Yasunaga et al., 2022) for drug synergy prediction. We modify the task input to be in natural language using a simple formulation:

$$[\text{CLS}] \, d^1 \, [\text{SEP}] \, d^2 \, [\text{SEP}] \, c \, [\text{SEP}]$$

where $d^1$ and $d^2$ are drug names (e.g., *imatinib*, *5-FU*), and $c$ is the name of a cell line (e.g., *MCF2*, *Ishikawa*). The model is then trained to predict the output value $y$ from the **[CLS]** token representation.

We also investigate to what extent pretraining knowledge is responsible for the model's performance. To do so, we evaluate the impact on performance when the drug and cell names are replaced with 'random' tokens. Given the ordered (by frequency) vocabulary $\mathcal{V}$ of the LM, we select the tokens $\{v_i \in \mathcal{V} \mid i \in [k..(k + |C| + |\mathcal{D}|)]\}$ to represent our drug and cell lines. Note we start at a threshold $k$ to avoid the most common tokens which might have specialized representations in the language model's latent space. We uniquely map each cell line and drug to a token in this set, which we use as input to the BERT LM. Essentially, this experiment is used to determine whether knowledge from pretraining or the transformer architecture itself is responsible for performance on the drug synergy task. An example input from this strategy is: [CLS] rabbit [SEP] fish [SEP] book [SEP].

### 3.2 SynerGPT: In-Context Learning for Few-Shot Synergy Prediction

#### 3.2.1 In-Context Learning for Function Classes: Background

Recent work trained transformer models to "in-context learn" function classes (Garg et al., 2022). A function class is a set of functions that satisfy specific properties, such as linear functions or neural networks. In-context learning of a function class $\mathcal{F}$ is defined as being able to approximate $f(x_{\text{query}})$ for "most" functions $f \in \mathcal{F}$ given a new query $x_{\text{query}}$ when conditioned on a prompt sequence $(x_1, f(x_1), \ldots, x_n, f(x_n), x_{\text{query}})$. We define a prompt prefix $P^n := (x_1, f(x_1), \ldots, x_n, f(x_n), x_{n+1})$ as the first $n$ in-context examples followed by the $n + 1$th input. A model $M_\theta$ parameterized by $\theta$ is trained to minimize the loss averaged over all prefixes

$$\min_\theta \mathbb{E}\left[\frac{1}{n+1} \sum_{i=0}^{n} w_i \ell\left(M_\theta(P^i), f(x_{i+1})\right)\right] \tag{1}$$

given some appropriate loss function $\ell$. Weights $w_n := 1$ unless otherwise noted.

#### 3.2.2 Predicting Drug Synergy In-Context

For in-context prediction of drug synergy, we redefine

$$P^n = (d_1^1, d_1^2, c_1, y_1, \ldots, d_n^1, d_n^2, c_n, y_n, d_{n+1}^1, d_{n+1}^2, c_{n+1})$$

as the prompt prefix (as discussed in Section 2, we refer to this as the "context" or "input context"). Here, $y$ can be considered the output of a function measuring synergy on $(d^1, d^2, c)$. As in (Garg

et al., 2022), we consider a GPT-2 family (Radford et al., 2019) decoder-only model, which we call SynerGPT. Here, the prediction of the synergy value $y_j$ is made using a linear transformation of the contextualized output representation of $c_j$ (note that this includes $d_j^1$ and $d_j^2$ due to self-attention). Model inputs–drugs $d$, cell lines $c$, and labels $y$–are initialized using a learnable embedding layer (i.e. no external features). To evaluate the model's ability to predict synergies of unknown entities, we hold out either $m$ drugs or $m$ cell lines and remove their synergy relations from the training set (see Section 4). We use a subset of the held out tuples as a pool of context examples. We now turn to selecting the context (prompt prefix) from this pool in a manner that increases predictive performance.

### 3.2.3 How to sample the context?

A central question about using language models without external features—including textual names—is how to teach the model to understand unknown drugs or cell lines. We propose using a masking strategy—every unknown drug $d^h$ or cell $c^h$ is represented by **[UNKNOWN]** and the model must use in-context learning to understand it based on contextually-related known drugs and cell lines. In this setting, we assume that we are given a set of synergy tuples to sample from to construct a prompt. During training, it's simply the training set. During evaluation, we consider a special held-out "context" set $\mathfrak{D}^c \subset \mathfrak{D}$ (thus named because we sample the context/prompt $P^n$ from this set). To sample from this context set, we propose a context-selection strategy based on constructing a graph $\mathcal{G}$ on this $\mathfrak{D}^c$. Specifically, we construct $\mathcal{G}$ by creating a node for every synergy tuple $x := (d^1, d^2, c, y) \in \mathfrak{D}^c$. We construct a drug edge $e^d$ between two nodes $x_1$ and $x_2$ if they share drug $d$ (i.e. $d \in x_1 \wedge d \in x_2$). Similarly, we construct a cell line edge $e^c$ if they share cell line $c$. See Figure 1 for an example and Appendix Figure 8 for more details. We employ the following context selection strategies to sample a context with $n$ examples given some node $x$ containing unknown $h$ which is either drug $d^h$ or cell $c^h$:

1. **Random**: Uniformly select $n$ context examples from $\mathfrak{D}^c$.

2. **Graph**: Uniformly select examples from the nodes adjacent to $x$ in $\mathcal{G}$.

3. **Unknown-First**: Uniformly select nodes adjacent to $x$ which share an edge of type $e^h$, i.e. prioritizing selection of nodes that contain the masked unknown $h$.

Note that these strategies are hierarchical– **Unknown-First** falls back to **Graph** when there aren't enough examples which falls back to **Random**. Examples from **Random** are put earlier in the context than **Graph** which is again put before **Unknown-First**. In order to train the model to correctly use the **[UNKNOWN]** token, we need to artificially create unknown drugs or cells during training. Given training example $x$, we uniformly select $d^1 \in x$ or $d^2 \in x$ to be the hidden drug $d^h$. For the unknown cell line setting, $c \in x$ is always set to $c^h$ because there is just one cell line per example. We replace all occurrences of $h$ in the prompt with **[UNKNOWN]**. We note that our sampling strategy is related to retrieval augmented models (Mialon et al., 2023). Here, however, we note that the model is also in-context learning synergy functions for unknown drugs based on Definition 1 in (Garg et al., 2022).

### 3.2.4 Optimizing the Context

We further study whether the context can be optimized to best enable predictions for some unknown drug or cell line $h$ (see Figure 7 for an example). The purpose of these experiments is to enable the eventual development of a standardized assay for drug synergy prediction. Thus, as output, these optimization algorithms produce a set of context tuples for each $h$. To do this optimization, we assume that we have four splits of data, which are constructed as follows. Given a set of $p$ "unknown" drugs/cells $H$, all synergy tuples not containing any $h \in H$ are put into a training set $\mathfrak{D}^{Tr}$. The remaining tuples are randomly partitioned into three equal sized sets: a context bank $\mathfrak{D}^c$, a validation set $\mathfrak{D}^v$, and a test set $\mathfrak{D}^{Te}$. We first train a model on $\mathfrak{D}^{Tr}$ following the **Unknown-First** strategy (where contexts are sampled from $\mathfrak{D}^{Tr}$ itself). Following this, for each unknown entity $h_i$, we select $n$ context examples from $\mathfrak{D}^c$ which maximize the model's score on the validation set $\mathfrak{D}^v$. This is a combinatorial optimization problem which can be considered related to the best subset selection problem (Bertsimas et al., 2015; Miller, 2002). We consider a genetic algorithm (Gad, 2021): a metaheuristic method which is useful for black box optimization of systems containing complex interacting parts (Mitchell et al., 2007), which is suitable for the complex interactions between cellular pathways required for drug synergy prediction. As output, we get a set of context tuples for each $h$. Optimization algorithm details are given in Appendix C.

### 3.3 In-Context Learning for Inverse Design

To train the model to retrieve relevant drug structures in-context, we use the same architecture as for synergy prediction (§ 3.2.2), so that we can use the same data split and optimized contexts from Section 3.2.4 to understand how the model interprets them. For effective retrieval, we need a strong base molecular representation that makes it possible to effectively distinguish molecules. So, we choose to use MegaMolBARTv2 (NVIDIA Corporation, 2022) representations, which were trained on 1.45 billion molecular SMILES strings and thus have a relatively comprehensive (in terms of drug classes) latent space. We train a SynerGPT model from scratch to predict representations using a linear transformation on the output **[UNKNOWN]** representation. We use this final representation to retrieve the desired drug using cosine similarity with the MegaMolBARTv2 representations of the drugs in our synergy dataset. The training context is selected using the **Unknown-First** strategy. Finally, we train the model using a minibatch contrastive loss (Radford et al., 2021; Edwards et al., 2021) between the L2-normalized ground truth representations $D^g$ (here MegaMolBartv2) and predicted representations $D^p$ (output from our model's prediction head):

$$\ell(D^g, D^p) = CE(e^\tau D^g D^{pT}, I_b) + CE(e^\tau D^p D^{gT}, I_b) \tag{2}$$

where $CE$ is categorical cross-entropy loss, $b$ is the mini-batch size, $I_b$ is the identity matrix, and $\tau$ is a learnable temperature parameter. We use this loss for $\ell$ in equation 1.

## 4 Results

**BERT can do Drug Synergy?** In this section, we experiment with finetuning BERT on drug synergy data where all drugs and cell lines are seen during training (data splits detailed in Appendix B.1). As discussed earlier, there has been recent work using external network datasets capturing interactions between drugs, proteins and cell lines (Yang et al., 2021) for synergy prediction. To evaluate the impact of these external datasets, we compare against a strong and recent model, Graphsynergy (Yang et al., 2021) that uses over a dozen different network datasets and achieves state-of-the-art on its subset of DrugCombDB.

We train four BERT-based (Devlin et al., 2019) language models (Beltagy et al., 2019; Yasunaga et al., 2022) and find that they outperform Graph-Synergy in both name and random token settings. BioLinkBERT with random tokens, for example, achieves a ROC-AUC score of 84.1% compared to GraphSynergy's 83.4% (p < 0.05 using paired *t*-test). In comparison, BioLinkBERT

| Model | KB | Name | ROC-AUC | PR-AUC |
|---|---|---|---|---|
| DeepSynergy | × | | 84.3 | 70.4 |
| MR-GNN | × | | 77.9 | 62.6 |
| SSI-DDI | × | | 63.3 | 41.4 |
| DeepDDS | × | | 87.2 | 77.0 |
| SciBERT (random) | | | 86.9 | 76.3 |
| BioLinkBERT (names) | | × | 86.4 | 75.9 |

Table 1: Classification results for four selected ChemicalX (Rozemberczki et al., 2022b) baselines and BERT on DrugCombDB (Liu et al., 2020). SciBERT and BioLinkBERT take random token and names as input, respectively. Values are average of five runs. Notably, SciBERT (random) outperforms four of the other five baselines. KB means external knowledge is used.

with drug names as input achieves 83.6%. We checked multiple BERT configurations, and details on other BERT models are shown in Appendix B.1 Table 4.

A natural question here is whether the model has learnt the required knowledge during pre-training. Surprisingly, replacing drug and cell names with random tokens (§ 3.1) resulted in no drop in performance. This suggests that the transformer architecture may be the dominant factor explaining BERT's performance on the task. However, if we use a randomly-initialized BERT model without any pre-training, we find the performance is worse (by 3 ROC-AUC pts).

To verify our findings, we consider the ChemicalX framework (Rozemberczki et al., 2022b), which implements several baselines and provides a standardized subset of DrugCombDB (Liu et al., 2020) with drug and cell line features. This standardization allows us to compare different baseline methodologies on the same dataset. The ChemicalX DrugCombDB dataset has 2,956 drugs, 112 cell lines, and 191,391 synergy tuples. We compare against baselines DeepSynergy (Preuer et al., 2018), MR-GNN (Xu et al., 2019), SSI-DDI (Nyamabo et al., 2021), and DeepDDS (Wang et al.,

2022), which we train using default hyperparameters from the original papers for 50 epochs as in (Rozemberczki et al., 2022b). These baselines (details in Appendix B.2) represent the most popular approaches to drug synergy prediction and allow us to compare against transformer architecture performance. Remarkably, SciBERT with *random* tokens outperforms all baselines except DeepDDS in this setting (Table 1). We see similar results on the DrugComb dataset (this database is larger but is continuously modified by volunteers; see Appendix L). We note that, while this performance is surprising in this domain, it follows from results from other domains. For example, language models are able to learn complex grammar and interactions just by observing how words co-occur. We conjecture this may be related to the observation that pre-training on a nonsense corpus (Krishna et al., 2021) can provide good weight initializiations for downstream tasks. We further discuss related work in Appendix F.

## 4.1 In-Context Learning for Few-Shot Drug Synergy

We now evaluate models on the few-shot and zero-shot setting, i.e, when a new drug or cell line is introduced with limited or no interaction data. We use the same architecture used in Garg et al. (2022): a GPT-2 (Radford et al., 2019) model with 256-dimensional embeddings, 12 layers, 4 attention heads, and batch size of 64. We use a learning rate of 2e-5. Model weights are initialized from scratch. To enable efficient experimentation in the few-shot setting, we construct a dataset split which contains multiple unknowns (i.e. $m$ held-out drugs or cells: $H := \{h_i \mid i \in [1..m]\}$). To construct our split, we remove all synergy tuples containing $h \in H$ from the dataset $\mathfrak{D}$ so that the remaining dataset only contains tuples with known drugs/cells (this is our training set $\mathfrak{D}^{Tr}$). Then, for each $h$, we select $n$ synergy tuples randomly to form the "context" bank/split $\mathfrak{D}^c$. All other "unknown" synergy tuples are put into $\mathfrak{D}^{Te}$.

For comparison, we use the same baselines trained in zero-shot and few-shot settings. We also test SetFit (Tunstall et al., 2022) (a few-shot LM approach), k-nearest neighbors, off-the-shelf pre-trained GPT-2 (using entity names as input, similar to CancerGPT (Li et al., 2023a)), and MAML with DeepDDS (details in Appendix B.2). In the few-shot setting, the context bank $\mathfrak{D}^c$ is considered part of the training set, and in the zero-shot setting it is not used. Our model, SynerGPT, however, is not trained on the context bank but uses it as context (prompt) examples for evaluation. Examples are selected using the Random, Graph, or Unknown-First strategies. We separately investigate the setting where drugs are unknown and where cell lines are unknown.

**Unknown Drugs** To construct the dataset split, we set $m = 50$ unknown, i.e.,"held-out" drugs and context $n = 20$ synergy tuples. Hence, our context bank contains $50 \times 20 = 1,000$ tuples. Overall, we find that our SynerGPT can perform better in the few-shot setting than existing baselines on on this task, as shown in Table 2. Full results are in Appendix Table 6. SynerGPT is trained in the zero-shot setting, which means it can be evaluated both with context examples (few-shot) and without any examples (zero-shot). Each strategy performs roughly the same zero-shot (although since strategies are used in training there are small differences), but the performance with sampled context examples is much different. Without examples, SynerGPT performs worse than DeepDDS few-shot, but the same SynerGPT model outperforms DeepDDS when given the few-shot context. Overall, we outperform all prior models in the few-shot setting and zero-shot setting. In particular, Unknown-First is able to increase performance by 3.8% absolute ROC-AUC with context, whereas DeepDDS only increases 1.3% from zero- to few-shot. Our approach is able to leverage the few given examples more effectively as shown by this higher increase in ROC-AUC. It is also notable that Unknown-First outperforms Graph since the context contains more examples with the unknown drug which the model is able to utilize to produce better predictions.

For example, the tuple (Vismodegib, Mithramycin A, NCI-H226) with unknown Vismodegib is True. Without examples, this is predicted as 0.46. For Graph with examples, it is predicted as 0.65–closer to the ground truth. For Unknown-First, the prediction further increases to 0.79. In this example, Graph only sees 15 examples containing the unknown but Unknown-First sees a full 20. Few-shot DeepDDS predicts 0.47 for this example, which is quite similar to our method without examples. As another example, (Chlorambucil, Cylocide, SK-OV-3) consists of two unknown drugs and has label False. Without examples, it is predicted as 0.62. Graph improves this to 0.35 and Unknown-First improves to 0.23. Interestingly, few-shot DeepDDS exhibits high uncertainty and predicts 0.50.

**Unknown Cell Lines** Since there are only 112 cell lines, we set $m = 20$ as unknown and use $n = 10$ context examples. Interestingly, we find that models perform worse with context examples. We

| Mode | Model | Unknown Drug | | Unknown Cell Line | |
|------|-------|------|------|------|------|
| | | ROC-AUC | PR-AUC | ROC-AUC | PR-AUC |
| Zero-Shot | DeepSynergy | 67.5 | 47.7 | 78.6 | 63.6 |
| | DeepDDS | 72.1 | 53.2 | 74.5 | 59.8 |
| | SciBERT (random) | 67.7 | 47.4 | 79.1 | 64.4 |
| | MAML-DeepDDS | 68.76 | 50.05 | 71.6 | 54.6 |
| | kNN-Features | 65.4 | 45.9 | 82.0 | 70.3 |
| | SynerGPT* (ours) | **74.0** | **57.3** | **83.5** | **72.1** |
| Few-Shot | DeepSynergy | 71.6 | 53.9 | 82.0 | 68.7 |
| | DeepDDS | 75.5 | 57.4 | 74.2 | 60.4 |
| | SciBERT (random) | 73.8 | 56.9 | 80.5 | 66.4 |
| | MAML-DeepDDS | 68.79 | 50.00 | 71.4 | 54.6 |
| | kNN-Features | 66.9 | 47.7 | 82.1 | 70.5 |
| | SetFit-S2 | 58.8 | 39.4 | 63.3 | 44.6 |
| | GPT-2 | 74.2 | 56.8 | 80.3 | 66.6 |
| | SynerGPT* (ours) | **77.7** | **61.5** | **83.8** | **72.8** |

Table 2: Few-shot and zero-shot results on ChemicalX DrugCombDB with 50 unknown drugs / 20 unknown cell lines. Our in-context methods perform better than baselines trained in the few-shot setting. Results are averaged over 5 runs. Zero-shot SynerGPT is evaluated without context. BERT models use random tokens. The difference between SynerGPT with and without context has p < 0.05 for both unknown drugs and cell lines based on a paired $t$-test. Similarly, both are statistically significant from the best baseline. *For simplicity, we report the best selection strategy (Unknown-First for unknown drug and Interpolate for unknown cell line). Full results are in Appendix B.2.

believe this is caused by the relatively small number of patient cell lines in the data vs. 2,956 drugs, making it harder to learn higher-level types of drug-cell line interaction. In other words, we are trying to learn a complex function class (drug synergy in an unknown cell line) without a significant number of example functions $f \in \mathcal{F}$. To alleviate this issue, we use 6 layers, batch size of 128, and only 30 epochs. Nonetheless, the issue still exists–performance decreases for baselines DeepDDS and MR-GNN and our strategies Unknown-First and Graph. We experiment with interpolating between training initially with Random to Unknown-First at the end (see Appendix B.2.2), which helps in the unknown cell line case. We believe this creates an exploration-exploitation effect.

## 4.2 CONTEXT OPTIMIZATION

As we have shown in the previous section that the context selection strategy is very important for SynerGPT performance, the natural next question is to what extent the context can affect model performance. To test this, we conduct a different split. Like before, we select 50 unknown drugs and 20 cell lines; with their respective tuples, we create three uniform splits: context, validation, and test. We train a SynerGPT Unknown-First model using hyperparameters as in our above experiments.

| Strategy | Unknown Drug | | Unknown Cell Line | |
|----------|------|------|------|------|
| | ROC-AUC | PR-AUC | ROC-AUC | PR-AUC |
| Mean UF | 79.2 | 63.8 | 85.2 | 74.9 |
| Best UF | 80.8 | 66.4 | 85.6 | 75.7 |
| GA | **81.5** | **66.9** | **86.1** | **76.5** |

Table 3: Test-set context optimization results ($p < 0.0001$). Model parameters are fixed, only context is changed. UF indicates Unknown-First strategy.

In context optimization, our goal is to select examples from the context and train splits which maximize some metric on the validation split. For our experiments, we maximize ROC-AUC for our trained model using the validation set. Overall, we consider two strategies: Unknown-First, and a genetic algorithm (GA). For the genetic algorithm, we use the implementation and hyperparameters from PyGAD (Gad, 2021) with a population of 8 for 50 epochs. Here, we consider each example in the context split to be a potential gene. For comparison, we also select the context at random according to the Unknown-First strategy. To ensure comparability, we evaluate Unknown-First the same number of times as the genetic algorithm and select the best context. Our results (Table 3) show that the genetic algorithm optimizes the context from a starting average AUC of 79.2% up to 81.5% for unknown drugs and from 85.2% to 86.1% for unknown cells. Appendix D visualizes this and shows error bars. We further analyze the results by different tissue types (Appendix G). For example, we find that for unknown drugs, synergy prediction in ovarian cancer is effective, but for both unknown drugs and cell lines predictive performance on bone cell lines is low.

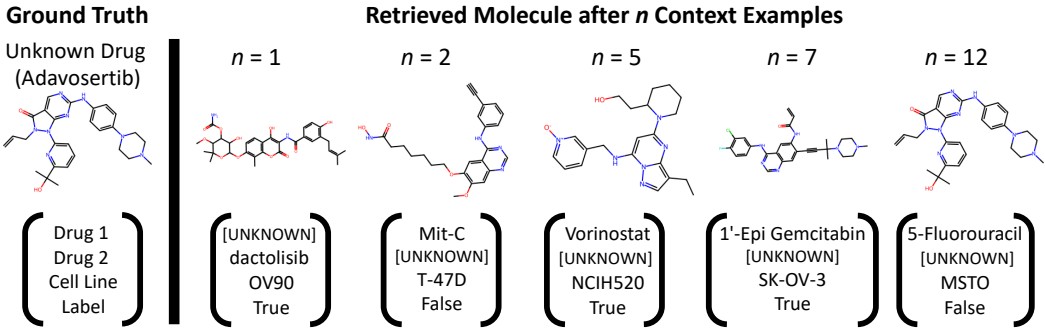

Figure 2: This figure shows the model's understanding of an unknown drug (Adavosertib) by retrieving candidates from the pool of held-out drugs. As the model sees more context synergy tuples $n$ (shown at the bottom; selected by the GA), it retrieves structures closer to the target molecule until finding the ground truth after 12 context examples. Repeated structures are skipped for brevity.

### 4.3 INVERSE DRUG DESIGN FROM SYNERGY EXAMPLES FOR DISCOVERY AND EXPLAINABILITY

Explainability is one of the most challenging problems in deep learning. With transformer language models, the contrast between remarkable performance gain and lack of explainability becomes even more striking. Here we propose a novel drug design task to better understand the model's "thought process." As shown in Figure 2, essentially, we look at SynerGPT's prediction as it gains more information via synergy tuples. While this is a useful step, we do recognize that retrieval doesn't fully address explainability and hope to inspire further work. We refer to Appendix A for more discussion.

In this novel task, we evaluate SynerGPT's ability to retrieve the structure of an unknown drug. We use the same splits as before but replace the classification head with a vector output trained using the loss in Equation 2. Using the same splits allows us to visualize the optimized context from the genetic algorithm. Experimentally, we achieve the best performance with the weight value from equation 1 set to $w_i := i/k$. Two examples of the model retrieving drugs which match the context synergy pairs are shown in Figures 2 and 5. These show the retrieved drug after $i$ context examples have been observed by the model. Additionally, we show overall retrieval performance as the number of context examples shown to the model increases in Appendix E, Figure 4. For the weighted strategy, mean rank for seen drugs decreases from ~1,500 to ~400 as context increases. Qualitatively, we find that we can retrieve the relevant drug or a similar structure from synergy relationships in multiple cases. This is considerably more effective for drugs observed during training, but performance is also better than random for unknown drugs. This ability to visualize the model's understanding is helpful for explaining what the model predicts from observing a given context. Second, it enables retrieving drugs which have a desired set of synergies, which can help inform drug candidate discovery, including patient-specific scenarios. We note that we worked off a broad definition of drug design as discovering new candidate medications. While retrieval is currently a challenging version of this, future work can expand the search space with generative models.

## 5 CONCLUSIONS AND FUTURE WORK

As demonstrated by HIV, HCV, and now cancer, combination therapy is a critical option for disease treatment. Yet, difficulties arise in regards to understanding drug-drug interactions and patient-specific genetic differences. To tackle this, we show that encoder-only language models are effective for drug synergy prediction. We then build on these results by proposing SynerGPT, a decoder model with a novel training strategy for in-context learning which can produce strong results for few-shot drug synergy prediction. We additionally show that the model context can be optimized using non-linear black-box approaches, which has exciting implications for the design of a standardized drug synergy testing panel for creating patient-specific synergy datasets. Finally, we explore a novel task of inverse design using desired drug synergy tuples. Performance on this challenging task is low for unknown drugs; nonetheless, it shows promise for future work that may enable personalized drug discovery.

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

# A    LIMITATIONS

While we are able to achieve strong performance without additional cellular or drug data, our approach is very much a black box akin to most deep learning methods. To address this, we propose the task of inverse design from drug synergy examples, which allows the visualization of the model's structural understanding as it gains more information. While this is a useful step, we do recognize that further research on mechanistic explainability would be valuable. We hope our contribution on synergy-based inverse design can inspire further work on explainability and that SynerGPT's predictions can be useful inspiration for clinical researchers. We would also like to note that regardless of using deep learning models, pharmaceutical researchers are in many cases unable to explain the mechanisms of many important drugs on their own (e.g., Modafinil, Metformin, general anesthetics) (Stahl, 2020; Rena et al., 2013; Brown et al., 2011) — let alone explain their interactions with each other. These drugs are prescribed to hundreds of millions of patients. Recent studies (Lin et al., 2019) suggest that many purported protein drug targets may not be the actual target at all. Important progress with life-saving modern drugs can be made with limited visibility into underlying mechanisms, yet certainly improved mechanistic understanding would be highly useful.

While we show that strong performance is possible without features, future work will still likely want to integrate external database features into drug synergy prediction; however, they will likely need to be integrated in a more thoughtful manner in order to ensure an actual benefit.

It would also likely be interesting for future work to investigate the internal connections language models are learning and what it might mean for understanding the fundamental biology of how cellular pathways interact. It is also worth noting that designing molecules using drug synergy tuples is a somewhat atypical task, so there may exist a wall in terms of the information content inherent in the context. While we do analysis by separating model performance into different tissue types in this work (as done in multiple prior studies), we note that for future research it is likely too limiting and simplistic to separate cell lines into tissues types.

# B    FULL RESULTS TABLES

## B.1    GRAPHSYNERGY FULL RESULTS

Full results for the BERT input method and GraphSynergy tests are in Table 4. We compare on the specific subset of DrugCombDB Liu et al. (2020) which was selected to match Graphsynergy's network data (i.e. selecting the subset of DrugCombDB with drugs/cells that can be matched with external protein-protein interaction, drug-protein association, and cell-protein association networks) and a 7:1:2 train:validation:test split. This data subset also contains useful surface names (the common natural language name of the drug; e.g. dasatinib), which allows us to compare the effect that drug names have on language model synergy prediction performance.

We consider three BERT training variations: the original BERT Devlin et al. (2019), SciBERT Beltagy et al. (2019), and BioLinkBERT Yasunaga et al. (2022). SciBERT was trained on a corpus of scientific documents which would be considerably more focused on drugs than a general corpus. BioLinkBERT is a biomedical BERT model additionally trained using document relation prediction (e.g. citation links). We would like to reiterate the rather remarkable finding that pre-training on scientific literature does not necessarily help the model perform drug synergy prediction any better. Overall, these results indicate that models using external data may not be behaving how we think they are.

**ChemicalX Results**    We report full results on the subset of DrugCombDB Liu et al. (2020) used by ChemicalX Rozemberczki et al. (2022b) in Table 5. Previous work tested on different subsets of existing datasets (due to filtering for external features).

| Input | Model | ROC-AUC | F1 | Precision | Recall | Accuracy |
|---|---|---|---|---|---|---|
| | GraphSynergy | 83.4 | 72.7 | 73.5 | 71.9 | 75.5 |
| Name | Unpretrained BERT-base | 80.6 | 71.0 | 71.7 | 70.3 | 74.0 |
| | BioLinkBERT-base | 83.6 | 73.1 | 73.4 | 72.8 | 75.7 |
| | SciBERT-base | 83.8 | 73.8 | 73.3 | 74.3 | 75.8 |
| | BERT-base | 83.8 | 73.3 | 74.2 | 72.4 | 76.1 |
| | BioLinkBERT-large | 84.7 | 73.9 | 74.7 | 73.1 | 76.7 |
| Random Token | BioLinkBERT-base | 84.1 | 73.7 | 73.6 | 73.8 | 76.2 |
| | SciBERT-base | 83.8 | 73.3 | 74.2 | 72.4 | 76.2 |
| | BERT-base | 84.0 | 73.4 | 74.1 | 72.7 | 76.1 |
| | BioLinkBERT-large | 84.1 | 73.8 | 73.4 | 74.2 | 76.1 |

Table 4: Performance of BERT models with names and random tokens and GraphSynergy on the custom subset of DrugCombDB Liu et al. (2020). Results are average of 5 runs. Name indicates that the common name of the drug is used as input, while Random Token uses the strategy described in Section 3.1.

| Model | KB Info | Name Info | ROC-AUC | PR-AUC |
|---|---|---|---|---|
| DeepSynergy | × | | 84.3 | 70.4 |
| MR-GNN | × | | 77.9 | 62.6 |
| SSI-DDI | × | | 63.3 | 41.4 |
| DeepDDS | × | | 87.2 | 77.0 |
| SciBERT (random) | | | 86.9 | 76.3 |
| BioLinkBERT (random) | | | 86.8 | 76.4 |
| BioLinkBERT (name) | | × | 86.4 | 75.9 |

Table 5: Classification results for four selected ChemicalX Rozemberczki et al. (2022b) baselines and two BERT-base models on DrugCombDB Liu et al. (2020). First two BERT models use random token inputs and last model uses drug names as input. Values are average of five runs.

## B.2 FEW-SHOT FULL RESULTS

### B.2.1 BASELINE DESCRIPTIONS

DeepSynergy is a popular feedforward model which uses cell line features and drug fingerprints. MR-GNN is a graph convolutional network (GCN) Kipf & Welling (2016) fed into an LSTM Hochreiter & Schmidhuber (1997) which takes the drug structure into account. SSI-DDI uses a graph attention network (GAT) Veličković et al. (2017) with a final co-attention layer. DeepDDS uses both a GAT and GCN, which are fed into a fully connected feed forward network.

**Real GPT-2** We train a GPT-2 model[2] in the few-shot setting (as opposed to SynerGPT's zero-shot) using random context and the same hyperparameters to mimic SynerGPT's training settings as much as possible. We use names of the drugs obtained from linking to PubChem Kim et al. (2023) as input in the form "Are drugs [DRUG1] and [DRUG2] synergistic in cell line [CELL]?".

**SetFit** Furthermore, we test finetuning a few-shot language-model baseline, SetFit Tunstall et al. (2022), on our few-shot data. We follow the original paper in using batch size 16, $R = 20$ text pairs generated for contrastive learning, and 1 epoch. Inputs to the model follow the same format as BERT in Section 3.1. We test using four models.

1. **SetFit-SBERT**: *paraphrase-multilingual-mpnet-base-v2* from Reimers & Gurevych (2019) with names as input. This model was trained to create semantic embeddings via Siamese networks.

---
[2] "gpt2" from HuggingFace.

2. **SetFit-C**: *recobo/chemical-bert-uncased-simcse* from Recobo.ai [3] with names as input. This model was trained using SimCSE on chemistry text.

3. **SetFit-S2**: *allenai/specter2* from Singh et al. (2022) with names as input. This model was trained on multiple scientific classification and regression tasks, such as MeSH descriptors classification.

4. **SetFit-SMILES**: *DeepChem/ChemBERTa-77M-MTR* from Ahmad et al. (2022) with SMILES strings as input. This model was pretrained by predicting 200 molecular properties for a molecule given its SMILES string.

**Model-Agnostic Meta-Learning**   We also consider a meta-learning formulation of our problem setting. We use MAML Finn et al. (2017) to train a DeepDDS model. Since MAML[4] does few-shot classification using episodes sampled from different learning tasks, we reframe our problem to match this. We consider predicting synergy for each drug to be a task. Then, we sample an episode for training from a random task for each mini-batch. We aggregate rare drugs without enough samples to form an episode into the same task until there are enough samples for an episode. Additionally, since we are dealing with binary classification here, we use $N = 2$-way. We sample the "validation" portion of each episode from our training set like in SynerGPT. We use the same context bank (and context size) for "adaptation" during evaluation. The same learning rate ($1e − 3$), batch size (512), and number of steps/epochs as DeepDDS is used. We report few-shot (first-order) and zero-shot (no adaptation) versions. Overall, we find that the MAML training procedure produces poor results, and adaptation produces insignificant performance increases. We attribute this to the episode-based sampling strategy neglecting important information in training.

**Protonets**   As another meta-learning baseline, we consider Protonets Snell et al. (2017). We use the same meta-learning framework as for MAML. Because we don't have drug task meta-data, we only consider the few-shot setting.

**k-Nearest Neighbors**   We also consider a k-Nearest Neighbors baseline using scikit-learn Pedregosa et al. (2011) similar to Nadkarni et al. (2021). We construct embeddings for each synergy pair by concatenating (Drug1, Drug2, Cell) embeddings. In the training set, we also include (Drug2, Drug1, Cell). We consider two embedding sources. For the first, kNN-Features, we consider the drug and cell fingerprint features from ChemicalX. For the second, kNN-S2, we use name embeddings from the Specter2 model. We report both zero-shot and few-shot versions. In the few-shot setting, the context bank is added to the training data. We set $k$ equal to the context number (20 and 10 for drugs and cell lines, respectively). We find performance on cell lines to be surprisingly effective, although still less than SynerGPT.

### B.2.2   Interpolate Details

In the Unknown cell line setting, we observe that Random has an interesting effect where it performs better after examples (although still worse than Unknown-First (no-ex)), so we consider a fourth strategy: interpolating between Random Unknown-First. Essentially, for each data mini-batch in epoch $e$ of $E$ total epochs, we select either the Random strategy with probability $\max(0.25, 1 − \frac{e}{E})$ otherwise we use the Unknown-First Strategy. This is analogous to an exploration-exploitation approach where we are pretraining with Random and transitioning to Unknown-First. We use a threshold of 25% to ensure the benefits of Random are kept until the end of training. We find that this interpolation strategy is effective (with $p < 0.05$, see Table 6) in dealing with the unknown cell line case.

### B.2.3   In-Context Implementation Details

In the unknown drug setting, to allow for tuples with multiple unknown drugs, we use both a **[UNKNOWN]** and **[UNKNOWN2]** token (e.g. a tuple containing two unknown drugs would be (**[UNKNOWN]**, **[UNKNOWN2]**, $c$)).

---

[3] www.recobo.ai

[4] We use the implementation from https://github.com/cnguyen10/few_shot_meta_learning

| Mode | Model | Unknown Drug | | Unknown Cell Line | |
|---|---|---|---|---|---|
| | | ROC-AUC | PR-AUC | ROC-AUC | PR-AUC |
| Zero-Shot | DeepSynergy | 67.5 | 47.7 | 78.6 | 63.6 |
| | MR-GNN | 65.9 | 44.7 | 76.6 | 61.9 |
| | SSI-DDI | 61.8 | 38.9 | 66.6 | 46.7 |
| | DeepDDS | 72.1 | 53.2 | 74.5 | 59.8 |
| | SciBERT | 67.7 | 47.4 | 79.1 | 64.4 |
| | BioLinkBERT | 65.8 | 45.6 | 79.0 | 64.5 |
| | MAML-DeepDDS | 68.76 | 50.05 | 71.6 | 54.6 |
| | kNN-Features | 65.4 | 45.9 | 82.0 | 70.3 |
| | kNN-S2 | 69.2 | 49.0 | 78.8 | 66.0 |
| Few-Shot | DeepSynergy | 71.6 | 53.9 | 82.0 | 68.7 |
| | MR-GNN | 68.1 | 48.4 | 76.5 | 62.1 |
| | SSI-DDI | 62.8 | 40.5 | 66.2 | 45.6 |
| | DeepDDS | 75.5 | 57.4 | 74.2 | 60.4 |
| | SciBERT | 73.8 | 56.9 | 80.5 | 66.4 |
| | BioLinkBERT | 73.0 | 55.6 | 80.6 | 67.4 |
| | GPT-2 | 74.2 | 56.8 | 80.3 | 66.6 |
| | SetFit-SBERT | 61.4 | 40.7 | 63.6 | 44.0 |
| | SetFit-C | 58.9 | 39.6 | 63.8 | 44.8 |
| | SetFit-S2 | 58.8 | 39.4 | 63.3 | 44.6 |
| | SetFit-SMILES | 63.6 | 43.6 | 64.6 | 44.5 |
| | MAML-DeepDDS | 68.79 | 50.00 | 71.4 | 54.6 |
| | Protonets-DeepDDS | 54.5 | 31.1 | 57.2 | 34.3 |
| | kNN-Features | 66.9 | 47.7 | 82.1 | 70.5 |
| | kNN-S2 | 70.0 | 49.9 | 79.0 | 66.2 |
| SynerGPT | Features | 73.4 | 55.5 | 70.7 | 52.7 |
| | Random (no-ex) | 72.2 | 54.5 | 77.1 | 61.3 |
| | Random | 73.7 | 56.8 | 82.3 | 70.2 |
| | Graph (no-ex) | 73.2 | 56.1 | 83.3 | 71.7 |
| | Graph | 75.5 | 59.6 | 83.2 | 71.5 |
| | Unknown-First (no-ex) | 74.0 | 57.3 | 82.9 | 71.1 |
| | Unknown-First | **77.7** | **61.5** | 81.7 | 69.9 |
| | Interpolate (no-ex) | | | 83.5 | 72.1 |
| | Interpolate | | | **83.8** | **72.8** |

Table 6: Few-shot and zero-shot results on ChemicalX DrugCombDB subset with 50 unknown drugs (left) and 20 unknown cell lines (right). Results are the average of 5 runs. no-ex indicates that our trained SynerGPT models were evaluated without any context examples. Features is a SynerGPT model where drug and cell line features are used instead of a randomly-initialized embedding layer. BERT models use random token inputs. Results are the average of 5 runs. The difference between Unknown-First with and without context has $p < 0.05$ for unknown drugs based on a paired $t$-test. On unknown cell lines using the interpolate strategy, $p < 0.05$. Similarly, both are statistically significant from the best baseline.

For the inverse design experiments, in some cases, context examples do not contain the unknown entity $h$ and therefore no [**UNKNOWN**] tokens, so we use $\overrightarrow{0}$ as a replacement for the ground truth representation when calculating our loss function. We use the same model, splits, and training hyperparameters as in the context optimization setting.

# C    CONTEXT OPTIMIZATION

## C.1    GENETIC ALGORITHM

In the case of the genetic algorithm, each context bank synergy tuple $x \in \mathfrak{D}^c$ is considered as a gene which can be selected by the algorithm. Given $p$ "unknown" drugs or cell lines, each has $n$ slots for context examples in its prompt, which makes for $np$ total genes. We also enforce that each $x$ contains the relevant unknown drug $d^h$ or cell line $c^h$. We disallow each context example from being selected multiple times; the reasons for this is two-fold. First, in early experiments we found that if we use the same example for the entire context (e.g. 20 repeats of $x$), then the model performs poorly. This is likely because the model is not trained on duplicate input, so it is trying to make meaningless connections between the same $x$. Second, repeating $x$ in the context provides no new information to the model. Although we enforce this constraint, in practice without it the model will likely do the same thing on its own.

For the genetic algorithm in context optimization, we use a population of 8 for 50 epochs. We use steady-state parent selection with 4 parents, single-point cross-over, 10% gene mutation, and elitism. Each example in the context bank is considered a gene and we disallow repeated genes. This results in 351 evaluations on the validation set.

## C.2    ERROR REDUCTION FOR CONTEXT OPTIMIZATION

Using the Unknown-First strategy, we sample a context for some heldout tuple in the validation set. We then calculate the absolute error $\epsilon$ for the heldout tuple. For each context example $x^c$ in the heldout tuple's input context $P^n$, we store $\epsilon$ and the relevant heldout entity, $h$. After some number (for fairness we use the same number of times as the genetic algorithm evaluates ROC-AUC on the validation set–351) of epochs on the validation set, we calculate a mean error $\hat{\epsilon}_h(x^c)$ for that context example $x^c$. Finally, for each heldout drug or cell $h$, we select the $n$ context examples $x_i^c$ with the lowest $\hat{\epsilon}_h(x_i^c)$.

As shown in Table 7, this strategy produces poor performance. This indicates that simply selecting all of the most individually informative context examples is not useful. Rather, there is a more complex, non-linear interaction between examples which is informative to the model. This is intuitive, because the interaction between cellular pathways in complex and still not well understood. The ability for the context to be optimized by a genetic algorithm but not error reduction indicates that data collection strategies which emphasize diversity may be important to consider for constructing new drug synergy datasets.

| Strategy | Unknown Drug | | Unknown Cell Line | |
| --- | --- | --- | --- | --- |
| | ROC-AUC | PR-AUC | ROC-AUC | PR-AUC |
| Typical Unknown-First | 79.2 | 63.8 | 85.2 | 74.9 |
| Best Unknown-First | 80.8 | 66.4 | 85.6 | 75.7 |
| Error Reduction | 75.4 | 59.0 | 84.9 | 74.5 |
| Genetic Algorithm | **81.5** | **66.9** | **86.1** | **76.5** |

Table 7: Test-set performance of different context optimization methods applied to the Unknown-First strategy. Note that the same model parameters are used in all cases and only the input context is changed. Considering Unknown-First as the distribution being sampled from, the genetic algorithm solution has a z-score of 4.02 indicating $p < 0.0001$.

## D    GENETIC ALGORITHM PERFORMANCE

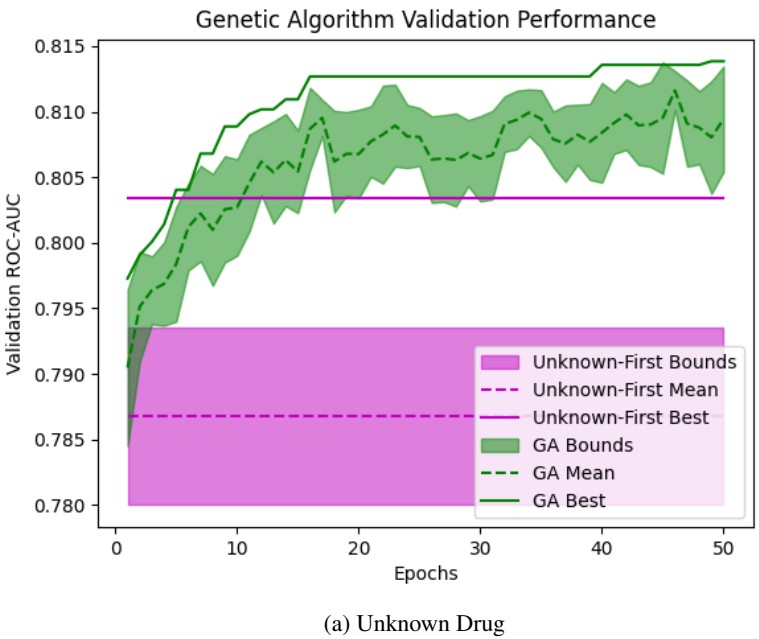

(a) Unknown Drug

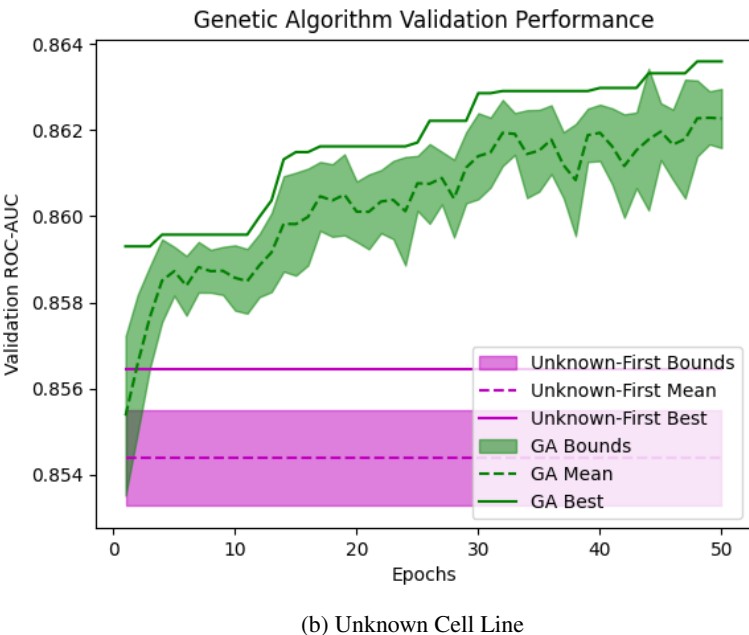

(b) Unknown Cell Line

Figure 3: Context optimization performance increase over epochs for the genetic algorithm on both unknown drugs and unknown cell lines. Highlighted areas shows error bounds. Purple regions shows an equivalent number of Unknown-First tests as the genetic algorithm.

## E    INVERSE DESIGN ADDITIONAL RESULTS

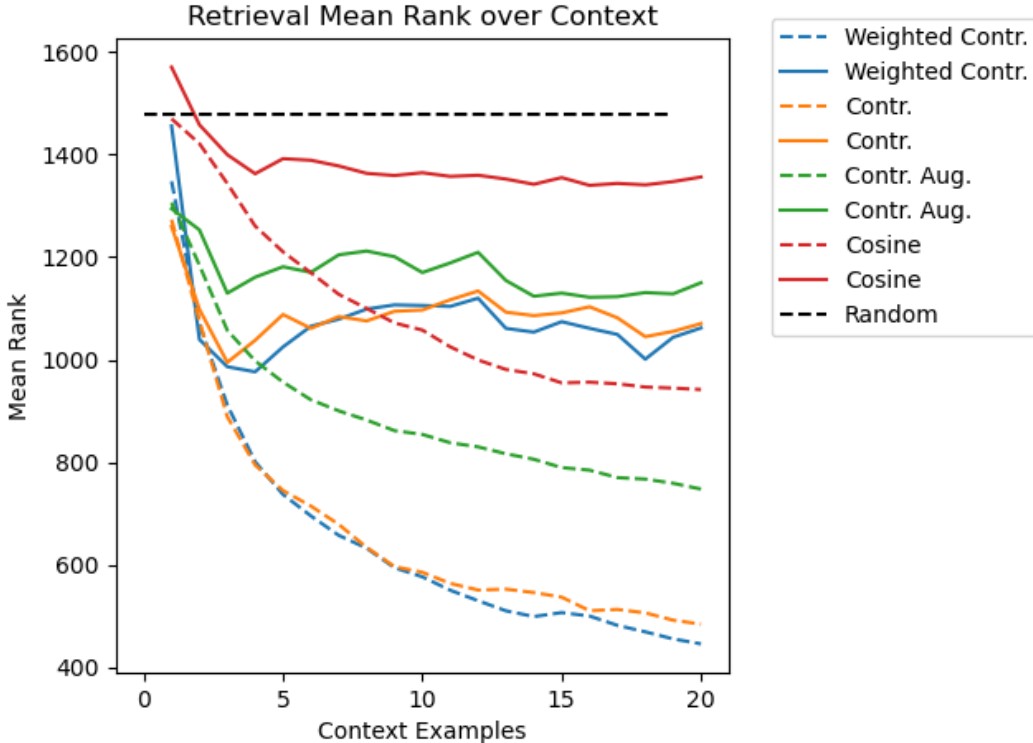

Figure 4: This figure shows the mean rank of the retrieved structure as more context synergy tuples are shown to the model. Solid lines show unknown drugs and dotted lines indicate known drugs. Four training variants are shown: Contr. is the contrastive loss described in Equation 2, Aug. uses five MegaMolBART representations from augmented SMILES strings for each drug. Cosine uses a simple cosine distance loss. Averaged over 5 runs. See 4.3 for details on weighted.

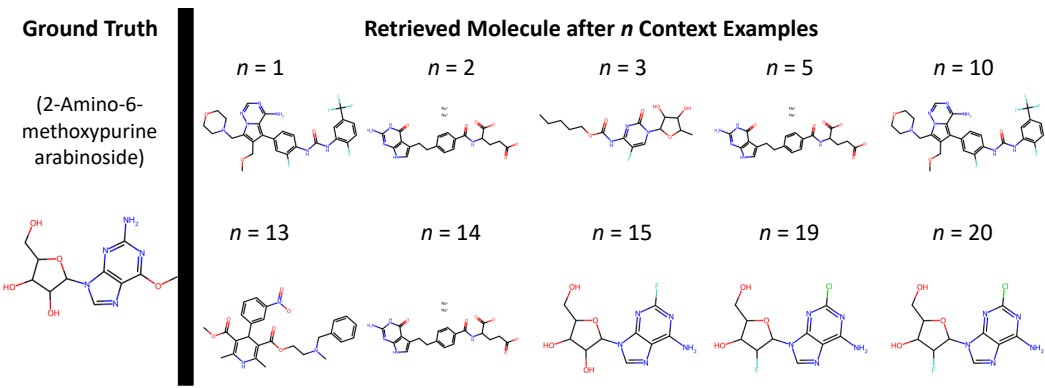

Figure 5: This figure shows the model's understanding of a known drug as it observes more context synergy tuples. The ground truth drug is not retrieved, but the model instead identifies two almost structurally identical drugs which would potentially be candidates for interacting with the same drug target. This shows how our method can identify related drugs which behave in similar, synergistic ways.

# F RELATED WORK

## F.1 MOLECULAR LANGUAGE MODELS

In recent years, advances in machine learning and NLP have been applied to molecule representations. Several efforts (Fabian et al., 2020; Chithrananda et al., 2020; Vaucher et al., 2021; Schwaller et al., 2021; NVIDIA Corporation, 2022; Tysinger et al., 2023) show excellent results training on string representations of molecules (Weininger, 1988; Weininger et al., 1989; Krenn et al., 2020; Cheng et al., 2023). Interest has also grown in multi-modal models (Edwards et al., 2022; Zeng et al., 2022) and multi-encoder models (Edwards et al., 2021; Vall et al., 2021; Xu & Wang, 2022; Su et al., 2022; Liu et al., 2022; Seidl et al., 2023; Xu et al., 2023b; Zhao et al., 2023) with applications to chemistry and biology. Existing work (Edwards et al., 2022; Su et al., 2022; Xu et al., 2023a; Christofidellis et al., 2023) also builds on this to "translate" between these modalities, such as MolT5 (Edwards et al., 2022), which translates between molecules and language.

## F.2 IN-CONTEXT LEARNING

With the success of models such as GPT-3 (Brown et al., 2020) and GPT-4 (OpenAI, 2023), interest has grown in the theoretical properties of in-context learning. (Garg et al., 2022), which we follow in this work, investigates the ability of transformers to learn function classes. (Olsson et al., 2022) investigates whether in-context learning is related to specific "induction heads". (von Oswald et al., 2022) shows that transformers do in-context learning by gradient descent. (Li et al., 2023b) frames in-context learning as algorithm learning to investigate generalization on unseen tasks.

## F.3 LANGUAGE MODELS FOR CHEMISTRY AND KNOWLEDGE GRAPH COMPLETION

Very recently, considerable interest has grown in using language models, particularly GPT-4 (OpenAI, 2023), for uncovering chemical knowledge and molecular discovery (Hocky & White, 2022; White et al., 2022; Bran et al., 2023; Boiko et al., 2023; White et al., 2023; Castro Nascimento & Pimentel, 2023), including work in the few-shot setting (Ramos et al., 2023; Jablonka et al., 2023). CancerGPT (Li et al., 2023a), a related contemporaneous preprint, was recently released which explores a similar few-shot approach to drug-drug synergy prediction. It explores training literature-aware text-based GPT models on drug synergy data. The use of GPT models pretrained on massive textual corpora from the web also makes rigorous evaluation and comparison difficult. We believe our work is complementary, since we largely explore the transformer architecture without language and we consider in-context learning which they do not. We also consider extensions such as inverse design and context optimization. Due to the recency of (Li et al., 2023a), we leave additional comparisons beyond our real GPT2 baseline to future work. Applying language models to knowledge graphs has been investigated in the general (Yao et al., 2019; Kim et al., 2020; Youn & Tagkopoulos, 2022) and scientific domains (Nadkarni et al., 2021; Safavi et al., 2022). They can be considered similar to our tests of BERT language models applied to a drug synergy hypergraph (§ 4).

## F.4 DRUG SYNERGY PREDICTION

As discussed above, there are several approaches (Preuer et al., 2018; Xu et al., 2019; Nyamabo et al., 2021; Wang et al., 2022; Kuru et al., 2021; Sun et al., 2020; Rozemberczki et al., 2022b) which can predict synergy scores given cell line and drug features. There has also been interest in learning representations for these settings (Scherer et al., 2022). Recently, work (Yang et al., 2021; Rozemberczki et al., 2022a; Lin et al., 2022) has begun to incorporate additional data sources such as drug-protein interactions. This can help improve results, but it often requires creating a subset of the original synergy dataset which can bias results towards the proposed method. (Yang et al., 2023) extracts additional training data from the literature to improve synergy prediction results, which may relate to our results in Appendix H. Research also investigates the application of few-shot (Ma et al., 2021) and zero-shot (Huang et al., 2023) machine learning to drug response prediction–we extend this idea to drug synergy prediction. (Yang et al., 2020) and (Kuenzi et al., 2020) are related but have different focuses compared to our paper; neither compare against any other synergy baselines or do large-scale evaluation. (Yang et al., 2020) focuses on a mechanistic understanding of (drug, tumor) activity–a different task. They use this understanding to rank subsystems and predict a limited

number of drug combinations to evaluate. (Kuenzi et al., 2020) does database and experimental testing with small numbers of cell tissues and drugs.

## G    TISSUE TYPE ANALYSIS

We further analyze the results of context optimization by separating the results for unknown drugs into their effects on different tissue types. To obtain tissue types, we use the COSMIC Tate et al. (2019) cancer mutation database. Results (Table 8) show that performance varies between different tissue types, but that the context optimized by genetic algorithm outperforms default Unknown-First and the model with no context in all cases with exception of pleura. For example, the model excels predicting synergies in ovarian cancers, but results are lower than average in bone and lymphoid cancers. For example, the ROC-AUC of ovarian cancer increases from 77.6 to 81.6% with examples selected using the default Unknown-First strategy. Our context optimization strategy also shows to be important– the ROC-AUC further increases significantly to 87.5 using the examples selected by the genetic algorithm. In the unknown cell line case, we see improvements on all cell lines except skin and bone. Interestingly, performance on bone-derived cell lines is low in both settings.

While we do analysis by separating model performance into different tissue types in this work (as done in multiple prior studies), we note that for future research it is likely too limiting and simplistic to separate cell lines into tissues types. Future studies may look at better bucketing approaches, such as primary cancer-driving mutations. An excellent example are KRAS mutations, which occurr in up to 25% of human tumors and in many different tissue types (for KRAS: pancreatic, thyroid, colorectal, and lung carcinomas, among others) Kranenburg (2005). Further, we note that while our work focuses on few-shot applications to mono-clonal cell lines and tumor biopsies, there is growing evidence that intra-tumor heterogeneity is a driving factor in cancer growth and is also responsible for drug resistance Black & McGranahan (2021). Future work can investigate the effect that this heterogeneity may have on patient-specific drug synergy prediction.

| Tissue Type | Genetic Algorithm | | No Context | | Typical Unknown-First | |
|---|---|---|---|---|---|---|
| | ROC-AUC | PR-AUC | ROC-AUC | PR-AUC | ROC-AUC | PR-AUC |
| skin | **84.1** | 71.1 | 77.6 | 61.0 | 81.6 | 67.8 |
| ovary | **87.5** | 80.0 | 81.2 | 73.1 | 86.1 | 77.7 |
| central nervous system | **82.1** | 66.6 | 76.4 | 54.3 | 79.2 | 61.6 |
| large intestine | **83.4** | 67.5 | 78.3 | 60.0 | 80.0 | 62.4 |
| pleura | 70.7 | 81.9 | **82.0** | 90.8 | 68.2 | 81.8 |
| haematopoietic and lymphoid | **74.1** | 60.5 | 67.9 | 52.7 | 73.6 | 58.2 |
| lung | **80.9** | 65.0 | 74.7 | 55.7 | 78.7 | 62.6 |
| bone | **69.6** | 65.5 | 69.2 | 65.7 | 68.3 | 65.2 |
| prostate | **82.6** | 67.1 | 78.5 | 61.0 | 79.5 | 60.6 |
| breast | **84.1** | 68.5 | 75.0 | 54.0 | 80.0 | 64.8 |
| kidney | **85.3** | 68.6 | 75.1 | 48.3 | 82.3 | 61.6 |

Table 8: Unknown drug synergy prediction results separated by tissue type. Results are shown without a context of examples, with the genetic algorithm optimized context, and with the default Unknown-First strategy.

| Tissue Type | Genetic Algorithm | | No Context | | Typical Unknown-First | |
|---|---|---|---|---|---|---|
| | ROC-AUC | PR-AUC | ROC-AUC | PR-AUC | ROC-AUC | PR-AUC |
| skin | 82.5 | 74.1 | 81.6 | 73.3 | **83.5** | 75.2 |
| ovary | **85.1** | 84.8 | 82.0 | 80.9 | 84.0 | 83.4 |
| large intestine | **88.6** | 81.4 | 87.6 | 79.7 | 87.9 | 80.3 |
| haematopoietic and lymphoid | **83.8** | 74.0 | 81.8 | 70.2 | 82.3 | 72.2 |
| lung | **84.2** | 67.2 | 82.9 | 66.9 | 83.2 | 65.3 |
| bone | 56.2 | 34.6 | 46.3 | 33.9 | **59.1** | 37.4 |
| breast | **89.3** | 80.7 | 88.9 | 79.9 | 88.8 | 79.8 |
| kidney | **85.3** | 70.3 | 84.2 | 68.0 | 84.5 | 69.2 |

Table 9: Unknown cell synergy prediction results separated by tissue type. Base model trained using interpolate strategy and evaluated using Unknown-First.

## H    HOW DOES TRAINING DATA SCALE WITH PERFORMANCE?

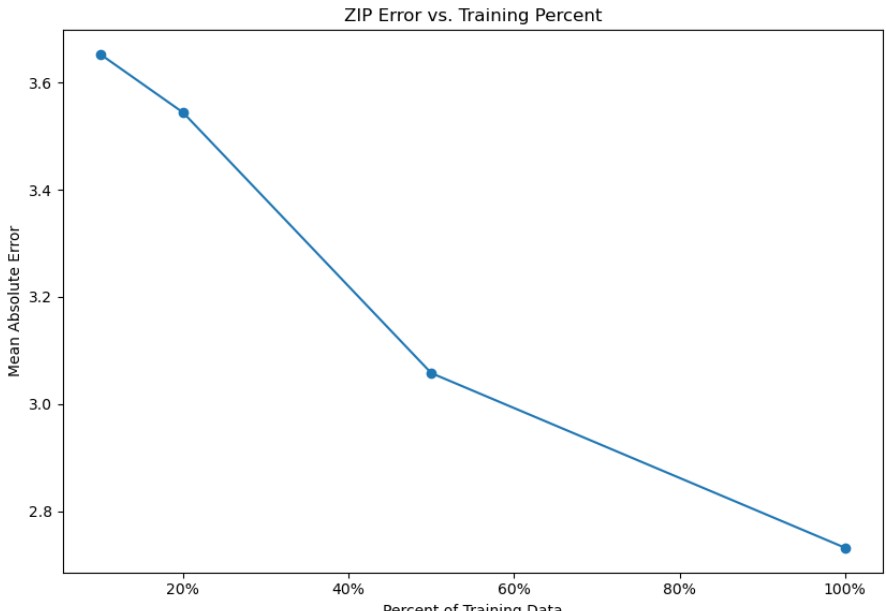

Figure 6: Performance versus percent of training data used for BERT with random token input.

Figure 6 shows how training data scale affects performance. We consider the version of the DrugComb Zagidullin et al. (2019) dataset for ZIP score regression recently released in the Therapeutic Data Commons software library Huang et al. (2021). It contains 129 drugs, 59 cell lines, and 297,098 synergy tuples. The figure shows BERT model validation performance trained using random token inputs.

## I    ADDITIONAL DIAGRAMS

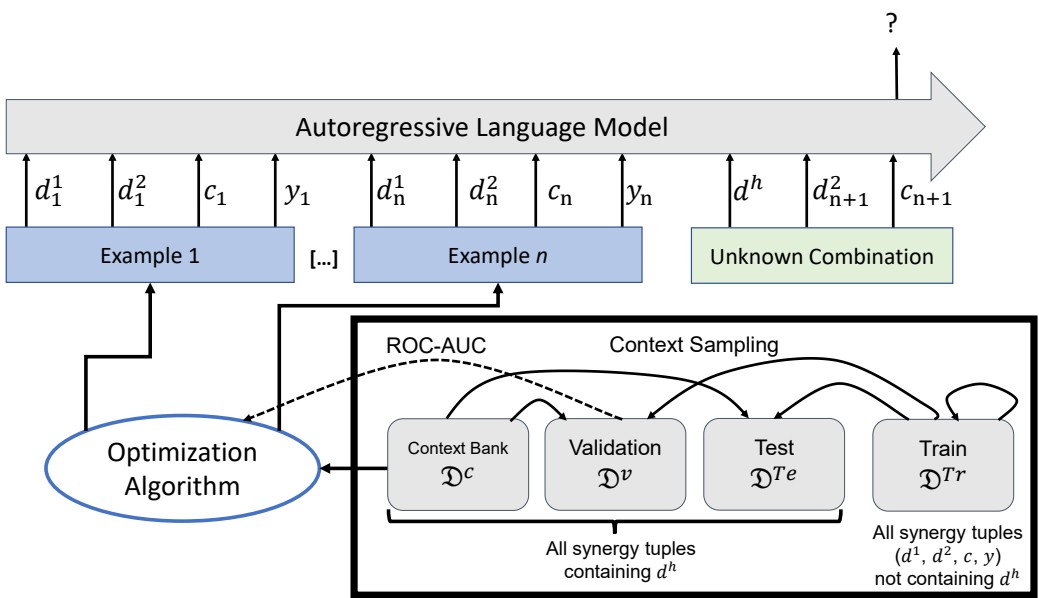

Figure 7: The process of optimizing the context (here showing the unknown drug setting). Solid arrows indicate where data can be accessed to build the model prompt. For example, evaluating the model on the test split can use synergy tuples from the train split and context bank. Training the model can only use tuples from the training split. The dotted line indicates that the model's evaluation on that split is used in the optimization algorithm. In our case, this is the ROC-AUC score on the validation set.

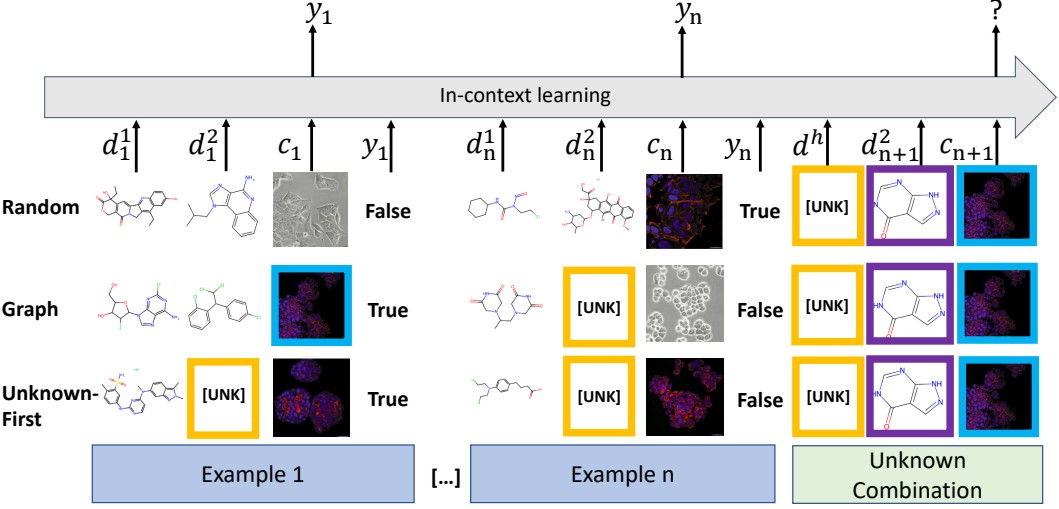

Figure 8: Example input of SynerGPT. Figure shows the three in-context learning strategies considered in this work. Colored boxes indicate when the input item is the same. Examples with the same colored box can be thought of as connected nodes in a graph. Example cell line images are from Stankevicius et al. (2016). Images are for visualization purposes– in practice a learnable embedding is used as described in 3.2.2.

## J    COMPUTING AND IMPLEMENTATION DETAILS

All experiments were done on an internal cluster of GPUs. Each experiment was conducted on a single NVIDIA RTX A6000 with 48 GB VRAM. Notably, multiple experiments can fit on the GPU at one time. Our BERT model experiments done within the ChemicalX framework take roughly 2.5

hours each. For SynerGPT, the Unknown-First and Graph variants took roughly 3 hours to train. Random was compute-bound by sampling, which caused it to take 9 hours to train. The inverse design variants took roughly 6-7 hours to train. We estimate that 80 days of GPU time were used for all experiments.

BERT-base consists of 108,233,473 parameters and BERT-large is 333,476,865. Unknown drug SynerGPT contains 22,793,473 parameters. Unknown cell version is 18,044,673 parameters. BERT-large models (we experimented with BioLinkBERT-large) were unstable to train in many cases. BERT and SynerGPT training used a linear decay learning rate schedule. Unknown drug SynerGPT uses 10,000 steps of warm-up. BERT used 1000 steps of warm-up. Unknown cell SynerGPT used 5% of training steps as warm-up. The training epoch for unknown drugs is 40 and unknown cell lines is 30. Since we used MegaMolBARTv2 embeddings, we used an output head size of 512 dimensions for retrieval. SynerGPT is trained using masked context examples selected from the training data– it does not see the unknown drug or cell line during training, which we categorize as zero-shot. Thus, when the SynerGPT model is evaluated on the test set without context examples, it is doing zero-shot prediction for the unknown drug or cell line. When in-context learning is done, it is few-shot. On ChemicalX DrugCombDB, we use a batch size of 512 with random tokens and 256 for names due to VRAM limits. For all random token BERT experiments, we use a high threshold of $k = 5,000$ to ensure no common tokens are used.

For the GraphSynergy dataset experiments, BERT-base models use a learning rate of 2e-5. We use 5e-6 for large models, which we find improves training stability. We use a batch size of 32.

Dataset split size varies depending on the seed for evaluating unknown drugs and cell lines. We detail our procedure for building these splits in Section 4.1 and 4.2. This is necessary because we conduct the split based on unknown drugs instead of as percentages. For Table 3 experiments, we follow the splitting procedure used in ChemicalX Rozemberczki et al. (2022b)– this yields on average (145766, 161647) training examples and (44625, 29544) test examples for unknown (drug, cell line), respectively. For the context optimization of unknown drugs and inverse design, the training set size is 145766 and the context, validation, and test set size are each 15208 examples on average.

## K EVALUATION METRICS

In this section, we detail the binary classification metrics that are used in this paper. Assume we have values for true positives $TP$, true negatives $TN$, false positives $FP$, and false negatives $FN$ where predictions are separated into positives and negatives based on some threshold $t$.

- Accuracy: (TP + TN) / (TP + TN + FP + FN)
- Precision: TP / (TP + FP)
- Recall: TP / (TP + FN)
- TPR: TP / (TP + FN)
- FPR: FP / (TN + FP)
- TNR: TN / (TN + FP)
- ROC-AUC Bradley (1997): The area under the curve created by plotting TPR against FPR as $t$ is varied.
- PR-AUC: Similar to ROC-AUC but the curve is TPR against Precision.
- $F_1$: 2TP / (2TP + FP + FN)

Given a list of rankings $R$,

$$MeanRank = \frac{1}{n} \sum_{i=1}^{n} R_i$$

## L    DRUGCOMB LANGUAGE MODEL EXPERIMENTS

In this section we consider whether our experiments on BERT hold on the DrugComb dataset Zagidullin et al. (2019); Zheng et al. (2021). Here, we use the version used in Rozemberczki et al. (2022b) which contains 4,146 drugs, 288 cell lines, and 659,333 synergy tuples.

| Model | KB Info | Name Info | ROC-AUC | PR-AUC |
|---|---|---|---|---|
| DeepSynergy | × | | 79.7 | 83.2 |
| MR-GNN | × | | 68.2 | 74.7 |
| SSI-DDI | × | | 55.8 | 62.7 |
| DeepDDS | × | | 81.7 | 85.8 |
| SciBERT (random) | | | 82.0 | 86.1 |
| BioLinkBERT (random) | | | 82.5 | 86.6 |

Table 10: Classification results for four selected ChemicalX Rozemberczki et al. (2022b) baselines and two BERT-base models on DrugComb Zagidullin et al. (2019); Zheng et al. (2021). BERT models use random token inputs. Values are average of five runs.

