# OpenReview forum: "SynerGPT: In-Context Learning for Personalized Drug Synergy Prediction and Drug Design"
_ICLR.cc/2024/Conference — ICLR 2024 Conference Withdrawn Submission_

### Official Review · Reviewer_UGB1 · 2023-10-28

**Soundness:** 2 fair
**Presentation:** 3 good
**Contribution:** 2 fair
**Rating:** 5
**Confidence:** 5

**Summary:**

This paper tackles an important problem, i.e. drug synergy prediction, especially in the case of few shot settings. This problem is challenging due to data scarcity. The authors leverage pretrained lanugage models to in context learn the task. Specifically, there is no external network knowledge injected, unseen drug and cell lines generalizations are reached by few shot, and the generative task of drug design is studied. The conclusions are confirmative that LLM could learn drug features without scientific context, few shot learning solves the small graph inference and drug discovery conditioning on patient data  is possible.

**Strengths:**

1. The paper is easy to follow and the presentation is clear
2. The in context learning method of LLM applied to synergistic prediction sounds novel
3. The context optimization study under a small graph is interesting

**Weaknesses:**

I'd thank the authors to explain on my questions below.

**Questions:**

1. Although I think the method of ICL is interesting, can the authors explain the difficulty behind the designing?
2. What is the motivation to choose encoder-only LM? How about encoder/decode LM?
3. I'd expect more analysis on the LLM being able to learn synergistic predictions with random tokens. What does the author think the learning come from?
4. Does the $\theta$ in eq (1) refer to the parameters of LM, either training from scratch or fine-tuning a model to optimize predictions given prompts?
5. I may misunderstand, why DeepSynergy has KB? Table 1
6. "However, if we use a randomly-initialized BERT model without any pre-training, we find the performance is worse (by 3 ROC-AUC pts)." Does it mean that you train from scratch a BERT?
7. Table 2, kNN-Features is comparable in unseen cell lines, but far lower in unseen drugs. Any analysis on this?
8. The baselines in Table 2 is bit old. BioLinkBERT which is used in table 1 is not compared here. Or could the author compare wrt other recent models?
9. I don't quite understand Figure 2. Contexts of which molecule are considered?
10. In the experiment with ChemicalX framework, how is the input controlled to be comparable? Which data preprocessing is used? For example, if I am not wrong, ChemicalX uses one-hot encoding for cell lines. Is this the same input for all baselines?
11. tiny. MAML-DeepDDS format in table 2

---

### Official Review · Reviewer_Af5r · 2023-10-28

**Soundness:** 3 good
**Presentation:** 3 good
**Contribution:** 2 fair
**Rating:** 3
**Confidence:** 5

**Summary:**

This paper is considered as one of the first adopting in-context learning framework of LLM for predicting synergistic drug combinations. Authors have followed the standard few shot learning framework including pretraining LLM with drug combination, prompt optimization, query, and evaluation. Authors also showed the Bert based discriminative classification gains no extra benefit by utilizing higher and more abstract text representation. Yet their decoder model shows competitive results. Authors also introduce another use case of inverse drug design for individual patients.

**Strengths:**

The paper presents a novel application of LLM based ICL in predicting synergistic drug combinations. Authors performed pre-training of a GPT model from scratch on known drug synergies, followed optimizing the in-context examples using genetic algorithm in the black-box setting, and systematically evaluated the performance in comparison with the traditional ChemicalX baselines and Bert based baselines. Authors also propose an interesting use case of inverse drug design.

**Weaknesses:**

The major weakness is lack of original contribution to LLM based In-Context Learning. The pre-training of GPT using known drug synergy pairs are a special case of the standard GPT pre-training. The in-context example selection is one using a heuristic algorithm without the gradient information, and evaluation are straightforward applications of ICL to a much simpler task. This is considered simple even in comparison with other basic tasks such as sentiment analysis, not to mention the more complex ones such chain-of-thought.

Findings from this paper are largely empirical and observational without foundation. For example, authors show that the text representation is not useful for their drug synergy prediction task by comparing their decoder-only model with Bert based model. It is not surprising since their decoder only GPT model is based on the next-token prediction, which does not use higher and more abstract textual representation anyway. Further, simplifying the in-context examples with all-single-token prompt (drug1-drug2-cell line-synergy) of course would improve performance particularly with only a few shots.

Authors also showed the text representation-based Bert approaches does not work as effective as their decoder only approach. It is also quite intuitive due to the simple all-single-token prompt task. A typical LLM typically allows a input of ~4000 tokens whereas this task only leverages a very small number of tokens as the prompt. Therefore, the so-called “intriguing questions” on non-textual pre-training could only be plausible for this simple task.

While the results are promising and use cases are interesting, this paper unfortunately does not contribute to our understanding of ICL by merely showing competitive performance on an overly simplified all-single-token task. As such, the reviewer believes, with added domain validation of the selected prediction of drug synergy, this work can be more suitable for publishing in a bioinformatics venue where the domain application and empirical impact are more appreciated.

**Questions:**

The reviewer doesn't have specific question for authors to address as this paper is considered as a specific domain application to cancer treatment. By presenting a specific use case or application, it does not increase our fundamental understanding nor knowledge of in-context learning in general.

---

### Official Review · Reviewer_DAPr · 2023-10-30

**Soundness:** 2 fair
**Presentation:** 1 poor
**Contribution:** 2 fair
**Rating:** 5
**Confidence:** 2

**Summary:**

The paper proposes to use pre-trained GPT language models to learn personalized drug synergy score prediction. The proposed method does not use any domain knowledge such as drug structures and protein interaction networks but purely from the names of the drug and cell line. They also propose a model that can do drug design which generates drugs that synergize and target a given patient.

**Strengths:**

The paper presents comparable results for the drug synergy prediction without using any drug or cell line chemiophysical information.

**Weaknesses:**

1. The writing style of the paper is a bit confusing and hard to follow.

**Questions:**

1. There are some parts of the paper that are a bit hard to follow, for instance:
In the paper, they claim that "Transformer LMs are Strong Drug Synergy Learners—Even Without Textual Representations" The textual representation refers to any text-based information that can be extracted from the drug and cell lines, but I believe the model eventually is used at least the name of the drugs which is also text?

 moreover, it is not clear
 "First, we consider drug synergy prediction using transformer language models without enriching
drugs/cells with information from external knowledge bases. We find these “feature-less” models are
able to achieve results that are better or competitive in comparison to knowledge-enhanced state-of-the-art drug synergy models"
if the author has trained a transformer language model here on the drug synergy dataset using drugs and cell line names as input or what does this model specifically refer to?

The following paragraph is also confusing to me
"We initially frame it as a retrieval task, effectively constraining the
output space, though from an implementation perspective, it is trivial to predict structures by
using a pre-trained generative model for molecules (Jin et al., 2020) with no architectural differences,
as both the retrieval and generation of drug structures require generating a latent vector."

The section for "In-Context Learning for Function Classes: Background" is also confusing, for instance, in equation 1, the expectation is taken with respect to which random variable? and how to understand $M_{\theta}(p^i) $

2. The SynerGPT was very shortly introduced in 3.2.2 which did not deliver a very clear description of the model, I am a bit confused on the model structure here, so it is a GPT-2 family decoder-only model, which is trained to predict y for a (drug, drug, cell) pairs, could you describe how to model structure looks like, why not encoder based model but decoder ?

3. Regarding section 3.3 I a not sure what you mean by:
" We train a SynerGPT model from scratch to predict representations using a linear
transformation on the output [UNKNOWN] representation. We use this final representation to retrieve
the desired drug using cosine similarity with the MegaMolBARTv2 representations of the drugs in
our synergy dataset. ", also in your inverse design,  can you generate new molecules or only retrieve from the existing dataset?

---

### Official Review · Reviewer_wxAg · 2023-11-01

**Soundness:** 2 fair
**Presentation:** 1 poor
**Contribution:** 2 fair
**Rating:** 3
**Confidence:** 4

**Summary:**

The paper proposes an approach for learning about drug synergy within a specific context. It operates with a personalized dataset of a few drug synergy relationships linked to particular cancer cell targets, with the objective of predicting additional synergies in that context.

Drawing inspiration from recent advancements in pre-training language models like GPT, which adapt to specific tasks, the paper develops innovative pre-training methods for enabling a GPT model to learn drug synergy functions within a given context. Notably, this model doesn't rely on textual data, molecular fingerprints, protein interactions, or other domain-specific knowledge but still achieves competitive results.

The paper incorporates an in-context approach with a genetic algorithm to optimize model prompts and identify potential drug synergy candidates. The intended outcome is to test these candidates following a patient biopsy. Moreover, the paper explores a new task called inverse drug design.

**Strengths:**

- The proposed SynerGPT obtains good performance, and provides a detailed theoretical analysis and experimental verification.

- The author attempts to propose a new and interesting task: Inverse Drug Design from Drug Synergy Context, and provides analysis and experiments on this task.

**Weaknesses:**

- The author claimed in sec 2 that it can handle the situation of multiple drug collaboration, but there is a lack of experimental proof of this part.

- For the new task proposed by the author, as the prediction can only target some properties of the drug, is there any non-uniqueness in predicting the drug results based on the synergistic effect of the drug? Does this task setting make sense? The authors define this task as a retrieval task to find results among existing drugs. But one of the main usage scenarios of this task should be new drug discovery. Is there any test of out-of-distribution generalization performance?

- The organization of experiment section is confusing. Why first evaluate BERT series on the task? As can be seen, they are not developed in this paper. If the target is to show that BERT series cannot handle the problem, the authors are suggested to put that part after showing the performance of SynerGPT.

- The authors did not compare with recent baseline [1].

[1] Few-Shot Drug Synergy Prediction With a Prior-Guided Hypernetwork Architecture, TPAMI 2023

**Questions:**

- In Section 3.2.3, would random sampling be useful? Ablation experiments should be done to prove it.

- Does combinatorial optimization result in more inference time? Experiment time consumption should be shown.

- Is it reasonable for the author to choose the representation of MegaMolBARTv2 as GT? Is MegaMolBARTv2 authoritative enough?


Please also reply to my questions in weakness.